# Indication of Light Stress in *Ficus elastica* Using Hyperspectral Imaging

**Pavel A. Dmitriev *** , **Boris L. Kozlovsky, Anastasiya A. Dmitrieva, Vladimir S. Lysenko, Vasily A. Chokheli** and **Tatyana V. Varduni**

Botanical Garden, Academy of Biology and Biotechnologies, Southern Federal University,
Rostov-on-Don 344006, Russia; blk@sfedu.ru (B.L.K.); admit@sfedu.ru (A.A.D.); vslysenko@sfedu.ru (V.S.L.);
vachokheli@sfedu.ru (V.A.C.); varduny@sfedu.ru (T.V.V.)
* Correspondence: pdmitriev@sfedu.ru

**Abstract:** Hyperspectral imaging techniques are widely used to remotely assess the vegetation and physiological condition of plants. Usually, such studies are carried out without taking into account the light history of the objects (for example, direct sunlight or light scattered by clouds), including light-stress conditions (photoinhibition). In addition, strong photoinhibitory lighting itself can cause stress. Until now, it is unknown how light history influences the physiologically meaningful spectral indices of reflected light. In the present work, shifts in the spectral reflectance characteristics of *Ficus elastica* leaves caused by 10 h exposure to photoinhibitory white LED light, 200 µmol photons $m^{-2} s^{-1}$ (light stress), and moderate natural light, 50 µmol photons $m^{-2} s^{-1}$ (shade) are compared to dark-adapted plants. Measurements were performed with a Cubert UHD-185 hyperspectral camera in discrete spectral bands centred on wavelengths from 450 to 950 nm with a 4 nm step. It was shown that light stress leads to an increase in reflection in the range of 522–594 nm and a decrease in reflection at 666–682 nm. The physiological causes of the observed spectral shifts are discussed. Based on empirical data, the light-stress index (LSI) = mean($R_{666:682}$)/mean($R_{552:594}$) was calculated and tested. The data obtained suggest the possibility of identifying plant light stress using spectral sensors that remotely fix passive reflection with the need to take light history into account when analysing hyperspectral data.

**Keywords:** hyperspectral imaging; fluorescence; photoinhibition; random forest; light stress; *Ficus elastica*





## 1. Introduction

Over the past decades, the frequency of extreme weather events during the growing season of plants, combining high light intensity and high temperatures, has increased dramatically [1]. Light stress occurs if the plant does not get rid of the excess energy received from the bright sun. Under stress from high light intensity, reaction centres become saturated, and excess energy can irreversibly damage photosystem II (PSII) [2]. This leads to photoinhibition—a steady decrease in the efficiency of photosynthesis [3]. Light stress during the growing season is usually combined with heat stress. Both of these stresses have a synergistic effect on plants [4,5].

One of the defence mechanisms against light stress is that plant leaves change the composition of pigments that absorb light. Another defence mechanism is that chloroplasts can move to shaded areas in plant cells. There is also a defence mechanism in which plants dissipate excess light energy as heat [6].

Light and heat stress lead to a significant decrease in plant productivity. The ability to quickly identify plant responses to stress is essential for improving crop-management practices and addressing the global food-security challenge. Therefore, an important task is to identify stress in plants quickly, using noninvasive and simple methods, over large areas.

Rapid diagnosis of light stress in plants over large areas is only possible using Earth remote-sensing (ERS) data. In confined areas, e.g., in greenhouses, foil tunnels, or open plantations, proximal spectral sensors can be used for stress recording. It is generally accepted that, among different remote-sensing techniques, the use of hyperspectral survey data is the most effective for plant-stress analysis [7]. A large number of remote hyperspectral studies were carried out on water-stress diagnostics [7–9]. Spectral sensors proved to be effective in diagnosing stress in plants from high [10–12] and low [13] temperatures. It is important to diagnose all kinds of stress before the first visible symptoms appear [14]. In this relation, there is a necessity to develop low-cost portable spectral optical sensors to accurately detect and diagnose the nature of plant stress [15].

The pigment composition of leaves, their photosynthetic characteristics and, accordingly, their optical properties are influenced by various stress factors, e.g., highly active forms of oxygen, air pollutants, drought, high temperatures, bright light, increased levels of UV radiation and others [16].

An evaluation of the physiological state of plants (including remote evaluations) may be performed using spectral characteristics and measurements of vegetation indices related to the contents of photosynthetic pigments [17–19]. In this regard, it is important to develop a new vegetation index (VI) for indicating plant stress.

The mechanism of the damaging effect of light excess on plants (photoinhibition) is mediated by the generation of superoxide ion $O_2{}^{*-}$ [20,21] and singlet oxygen $^1O_2$ [22] that can interact with photosystem I and II elements and impair their functions. In particular, the prerequisites for photoinhibition appear, if the electron light-excitation rate (i.e., number of P680 or P700 excited states generated per time unit) exceeds the electron transfer capacity of the thylakoid electron transport chain. In this case, there is a high probability of the transfer of excited electrons to $O_2$, followed by the formation of reactive oxygen species, which damage PSII and accessory pigments in antennae complexes [23], thus changing absorption spectra [21,22,24].

Thus, remote-sensing tools equipped with spectral sensors can identify stress in plants as such, but it is very difficult to determine its nature. Therefore, the development of a special technique for determining the light stress of plants using spectral sensors that record passive reflection is an important task.

The aim of the study was to determine the spectral ranges that make it possible to diagnose light stress in plants. The implementation of this task was carried out in laboratory conditions, in which the influence of random factors is minimized. The data obtained may be useful for the further development of simple and low-cost multispectral cameras applicable as Earth remote-sensing (ERS) tools.

## 2. Materials and Methods

*Ficus elastica* Roxb. ex Hornem plants were selected for the light-stress studies. They were expected to be sensitive to light stress, as they are short-day shade-tolerant plants. The plants were grown in the greenhouse of the Botanical Garden of the Southern Federal University under natural light (about 100 µmol photons $m^{-2}\,s^{-1}$), relative soil humidity of 60%, and a light/dark regime = 14/10 for 14 months. All the plants were of the same age.

Hyperspectral Imaging (HSI) was carried out using a Cubert UHD-185 [25,26]. The wavelength range of this hyperspectral camera is from 450 nm to 950 nm; the spectral resolution is 4 nm. The camera lens was located at a distance of 40 cm from the leaf blade of the plant and directed perpendicular to it (Figure 1). HSI was conducted under 4 halogen lamps and 1 blue LED. The integral spectrum of the measuring light is shown in Figure 2. Each hyperspectral image is presented as a single panchromatic image, $1000 \times 1000$ pixels in size and 125 spectral images, $50 \times 50$ pixels in size. The spatial resolution of the obtained hyperspectral data was approximately 35 $mm^2$.

Nine genetically homogeneous *Ficus elastica* plants, which were previously kept in the greenhouse conditions under natural light (Light conditions 0—LC 0), were placed in darkness for 14 h (LC 1). After exposure to LC 1, three plants were placed at a distance of

40 cm from the leaf under photoinhibitory light (Variant 1—Var. 1, LC 2, light stress) having a photosynthetic photon flux density (PPFD) of 200 μmol photons m$^{-2}$ s$^{-1}$. Three plants were placed under moderate light (Var. 2, LC 2, shade) (PPFD = 50 μmol photons m$^{-2}$ s$^{-1}$), and the remaining three plants were returned to the dark (Var. 3, LC 2). The levels of PPFD were measured using a PAR sensor of a Waltz Diving-PAM fluorometer. The duration of exposure at LC 2 was 10 h. After the experiment (Repetition 1—Rp 1), the plants were returned to the greenhouse. Two weeks later, the experiment was repeated (Rp 2). HSI was performed 4 times (HSI 1–HSI 4), before and after LC 2.

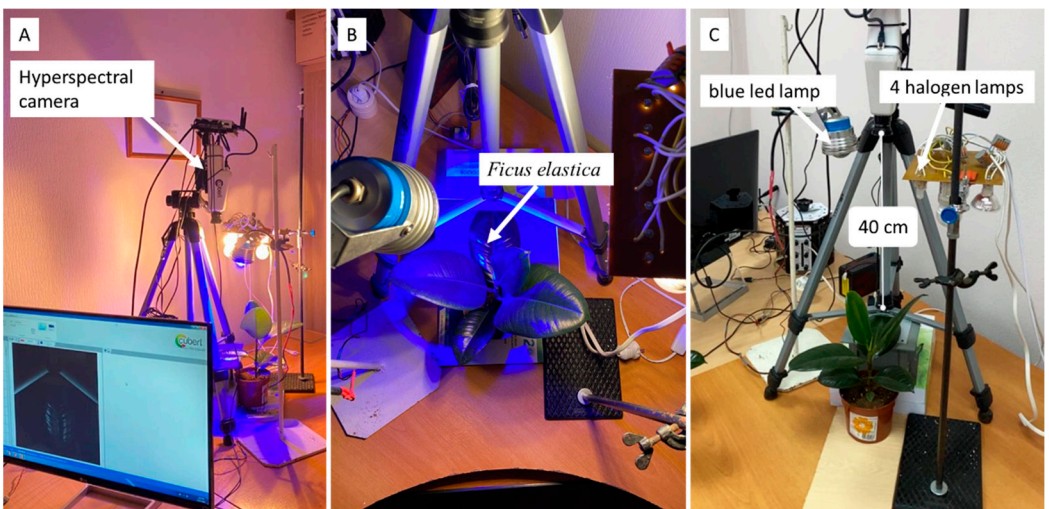

**Figure 1.** Hyperspectral imaging of *F. elastica* leaves. (**A**) Cubert UHD-185; (**B**) object of study; (**C**) artificial light sources for HSI.

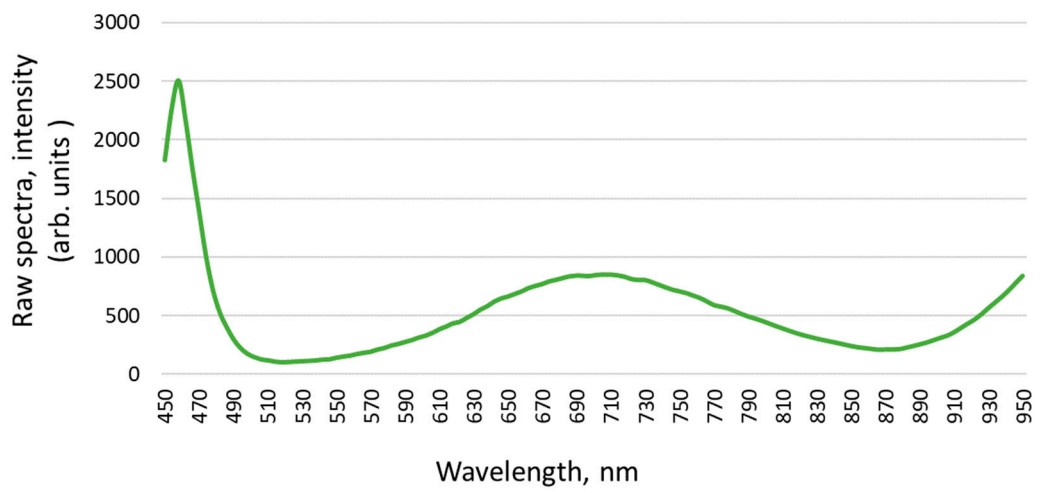

**Figure 2.** Integral spectrum of artificial light sources reflected from a white standard. Measured with a Cubert UHD 185 camera with a 4 nm step.

The third ficus leaf from the top was always used to obtain the hyperspectral data. The block diagram of the experiment is shown in Figure 3.

After HSI data preprocessing, 30 to 40 spectral profiles were randomly selected for each leaf from the top of the leaf blade since it is the flattest area which causes minimal distortion compared with the bottom and middle parts of the leaf blade. These bottom and middle parts are strongly curved symmetrically to the central vein.

The *t*-test and random forest (RF) were used to identify spectral ranges that allow for diagnosing light stress in plants. The choice of significant spectral ranges was carried out according to the values of the significance level of the *t*-test, out-of-bag (OOB) estimate of

the error rate, mean decrease accuracy, and mean decrease Gini according to the results of RF pixel-based classification.

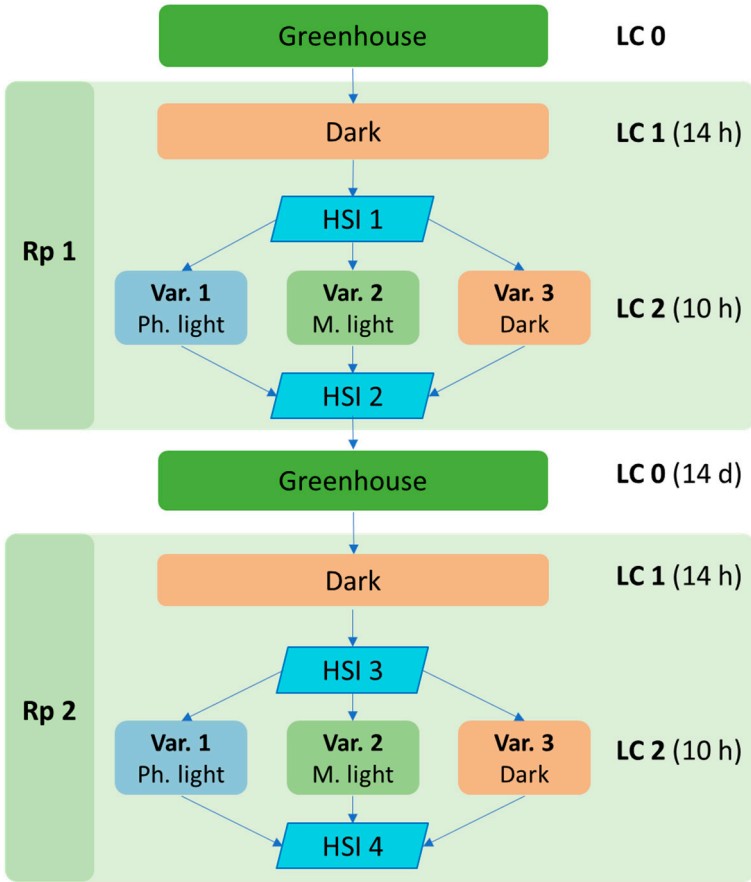

**Figure 3.** Block diagram of an experiment to study shifts in the spectral reflection characteristics of *F. elastica* leaves in the range of 450–950 nm. Var 1, light stress—10 h exposure to the photoinhibitory light (200 μmol photons m$^{-2}$ s$^{-1}$); Var. 2, shade—moderate light (50 μmol photons m$^{-2}$ s$^{-1}$); Var 3—dark-adapted plants. HSI 1–4—plant HSI; LC 0–2—light conditions; Rp 1–2—repetition of the experiment.

The experimentally calculated light stress index (LSI), was tested with RF against 125 spectral bands (SB) and 80 vegetation indices (VIs) for the mean decrease Gini and the mean decrease accuracy.

Statistical and mathematical analyses of the obtained results were carried out using the R environment [27].

Detailed information on these 80 VIs can be found in the work of Dmitriev et al. [28].

## 3. Results

### 3.1. Selection of the Most Informative SB for Identification of Light Stress

3.1.1. Selecting SB Using *t*-Test

To select the most informative spectral ranges, the values of the SB of the same ranges were compared between different lighting conditions (LC 2) using a *t*-test. The comparison results are presented in Supplementary Table S1 and Figure 4. For some compared bands, the level of significance (*p*) is lower than 0.0001. This did not allow for representing the dynamics of the significance level visually and proportionally in the form of a curve in the figure imposed over the entire spectrum. Therefore, the significance level values were converted to the negative decimal logarithm ($-\log_{10}p$). The dynamics of $-\log_{10}p$ over SB are shown in Figure 4.

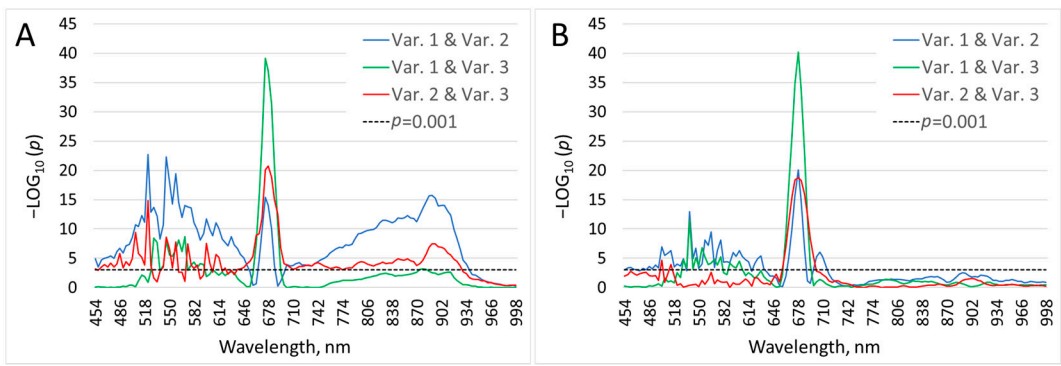

**Figure 4.** The $-\log_{10}p$ values, depending on the SB, when comparing variants (Var. 1, Var. 2, Var. 3; LC 2) by pairs. (**A**)—*t*-test (HSI 3); (**B**)—*t*-test (HSI 5).

The first zone in which significant differences are observed according to the variants in both repetitions of the experiment lies in the range of 498–610 nm. The peak of significance of the difference between the variants, in terms of the values of the SB, falls at 680–690 nm, i.e., the zone of chlorophyll fluorescence.

3.1.2. Selection of SBs by Their Contribution to Mean Decrease Accuracy and Mean Decrease Gini Based on the Results of RF Pixel-Based Classification

Next, the RF classification was used to select spectral ranges. It was applied to classify spectral profiles belonging to different lighting conditions (LC 2).

The SBs that have the greatest impact on Mean Decrease Accuracy and Mean Decrease Gini are shown in Figure 5.

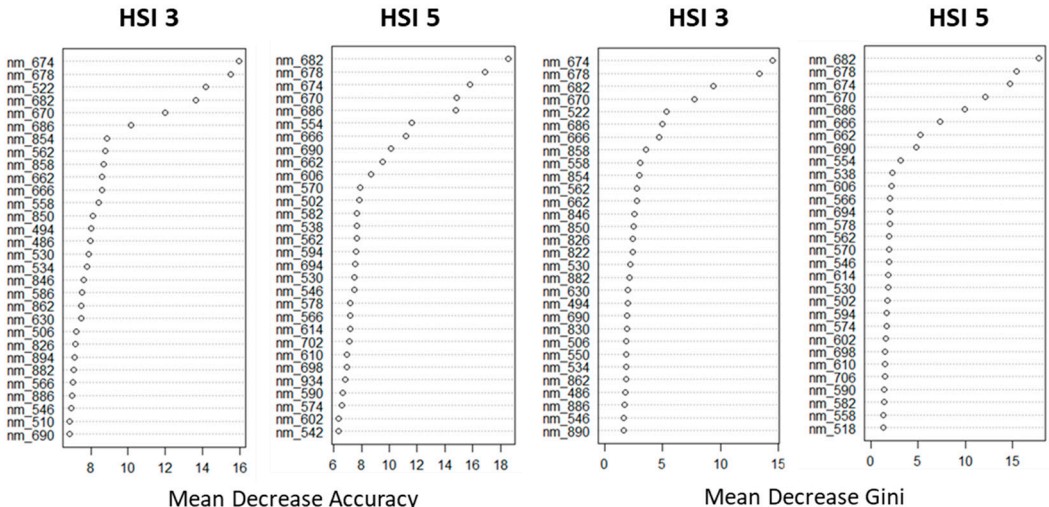

**Figure 5.** Ranking SBs by their contribution to mean decrease accuracy and mean decrease Gini.

Summarizing the results, the most important SB for mean decrease accuracy and mean decrease Gini for the two repetitions of the experiment lie in the ranges of 662–690 nm and 522–554 nm. Thus, in the process of identifying different lighting conditions, the RF pixel-based classification method revealed a significant effect of chlorophyll fluorescence in the red zone of the spectrum.

3.1.3. Selection of SBs by Searching for Their Optimal Combinations for RF Pixel-Based Classification

Spectral bands were selected according to their contribution to the OOB estimate of the error rate. A simple selection algorithm has been chosen that does not require significant computing resources. First, the pair of SBs that produced the smallest OOB

estimate of the error rate was selected. Next, a third band was selected for this SB pair, which, in combination with the first pair, gave the lowest error, and so on. The number of combinations of SBs (Y) is determined by the following equation:

$$Y = n \times (n-1) + \sum_{X=1}^{n-2} X \tag{1}$$

where n—number of bands.

The comparison was carried out for all pairs of variants (Var. 1 and Var. 2; Var. 1 and Var. 3; Var. 2 and Var. 3;) under different light conditions (LC 2) for HSI 3 and HSI 5. Spectral bands were ranked by their contribution to the OOB estimate of the error rate (Supplementary Table S2). Based on the pairs of SBs that make the greatest contribution to the decrease in the OOB estimate of the error rate, two spectral ranges were determined: 522–594 nm and 666–682 nm (Figure 6).

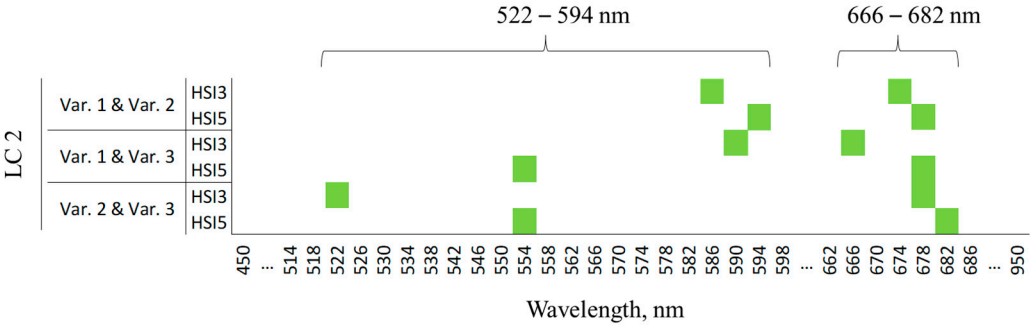

**Figure 6.** Pairs of SBs that make the greatest contribution to the decrease in the OOB estimate of the error rate.

*3.2. Comparison of a Simple Ratio of the Detected Spectral Ranges for the Identification of Light Stress*

Three variants of significant spectral ranges were selected:

- using a *t*-test (682–690 nm and 498–610 nm);
- by the contribution of SBs to the mean decrease accuracy and the mean decrease Gini according to the results of RF pixel-based classification (662–690 nm and 522–554 nm);
- by selecting the optimal combinations of SBs according to the value of the OOB estimate of the error rate of RF pixel-based classification (666–682 nm and 522–594 nm).

Further, based on the spectral ranges established for the identification of light stress, their simple ratios were calculated and tested.

$$\text{Light-stress index 1: LSI 1} = \text{mean}(R_{682:690})/\text{mean}(R_{498:610})$$

$$\text{Light-stress index 2: LSI 2} = \text{mean}(R_{662:690})/\text{mean}(R_{522:554})$$

$$\text{Light-stress index 3: LSI 3} = \text{mean}(R_{666:682})/\text{mean}(R_{522:594})$$

A comparison of boxplots of three variants (Var.1, Var.2, and Var.3) based on the results of HSI 3 and HSI 5 allows us to make a preliminary conclusion that LSI 3 is more informative for detecting light stress than LSI 1 and LSI 2 (Figure 7).

3.2.1. Comparison of LSI 1, LSI 2, and LSI 3 with SBs for the Identification of Light Stress

Classification of the state of ficus plants was carried out using RF pixel-based classification, according to values of the SB and LSI 1, LSI 2, and LSI 3. According to the classification results, LSI 3 has the largest contribution to the mean decrease accuracy and the mean decrease Gini values (Figure 8).

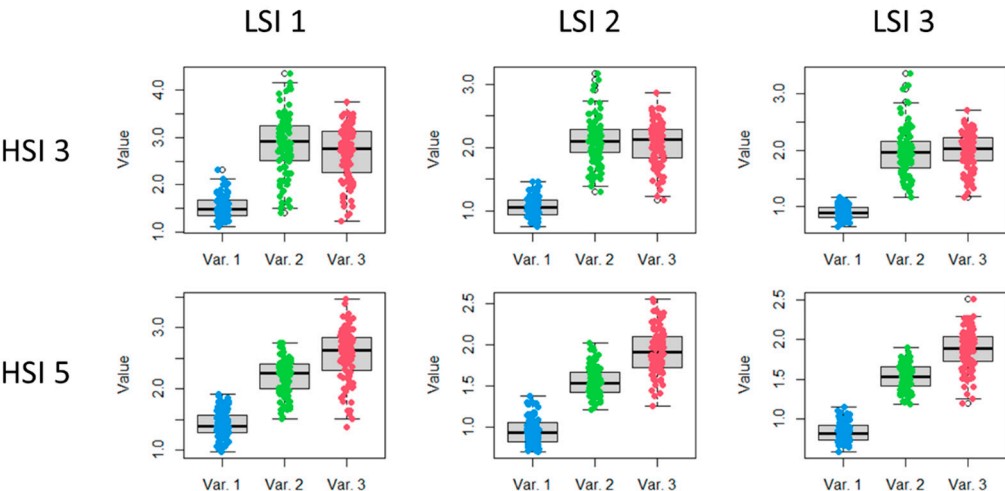

**Figure 7.** Boxplots of LSI 1, LSI 2, and LSI 3 values of three LC 2 variants (Var. 1, Var. 2, and Var. 3) based on the results of HSI 3 and HSI 5.

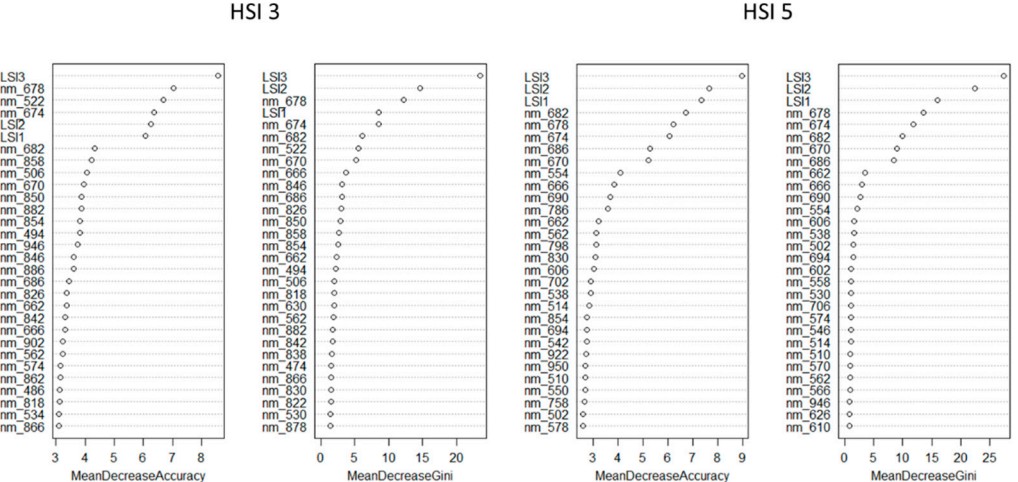

**Figure 8.** The most significant SB and LSI for indicating light stress in ficuses, depending on the mean decrease accuracy and the mean decrease Gini.

Boxplots of the LSI 3 values of three LC 2 variants (Var. 1, Var. 2, and Var. 3), according to the results of HSI 3 and HSI 5, compared to values of the SB 674 nm, 682 nm and 554 nm, are shown in Figure 9.

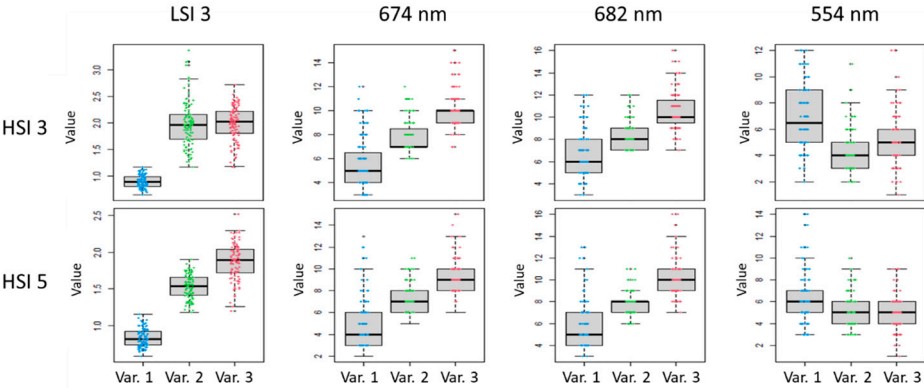

**Figure 9.** Boxplots of LSI 3 values compared to the values of the individual most significant SB.

### 3.2.2. Comparison of LSI 1, LSI 2, and LSI 3 with VIs for the Identification of Light Stress

When testing LSI 1, LSI 2, and LSI 3 values with 80 VIs using RF pixel-based classification, LSI 3 turned out to be the most significant depending on the mean decrease accuracy and the mean decrease Gini, as in the case of comparison with SB (Figure 10).

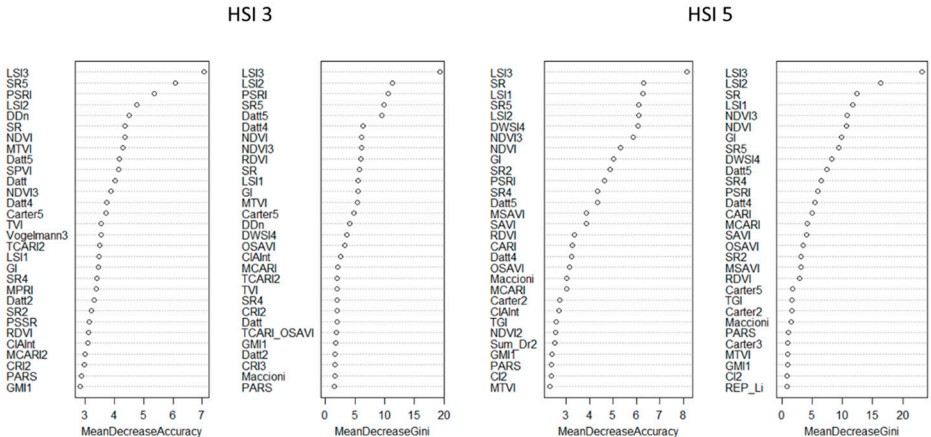

**Figure 10.** The most significant VIs for identifying ficus light stress, depending on the mean decrease accuracy and mean decrease Gini.

Boxplots of the LSI 3 values of three LC 2 variants (Var. 1, Var. 2, and Var. 3), based on the results of HSI 3 and HSI 5 compared to the values of VIs SR, SR5, PSRI and NDVI, are presented in Figure 11.

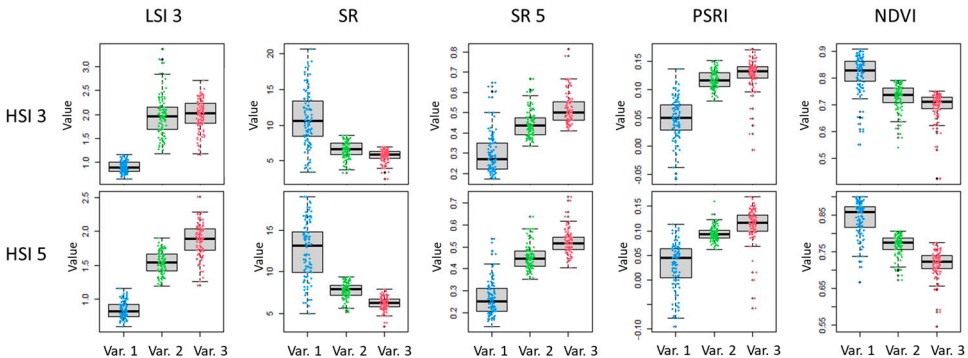

**Figure 11.** Boxplots of LSI 3 values compared to individual most significant VIs.

### 3.3. Light-Stress Index (LSI)

The LSI 3 values of the three LC 2 variants (Var. 1, Var. 2, and Var. 3), obtained according to the results of HSI 3 and HSI 5, were distributed under a normal law (Figure 12). This allows for the use of both parametric and nonparametric methods of statistical analysis when working with LSI 3.

It has been found that there is a linear relationship between the average values of the spectral ranges 666–682 nm and 522–594 nm (Figure 13). Therefore, a specific range of LSI 3 values may be applied to identify and evaluate the magnitude of light stress. It was preliminarily found that the LSI 3 threshold, which separates light stress from other plant states, is 1.17. Additional studies are needed to develop and verify the LSI 3 scale, which would allow for the light history of plants to be considered.

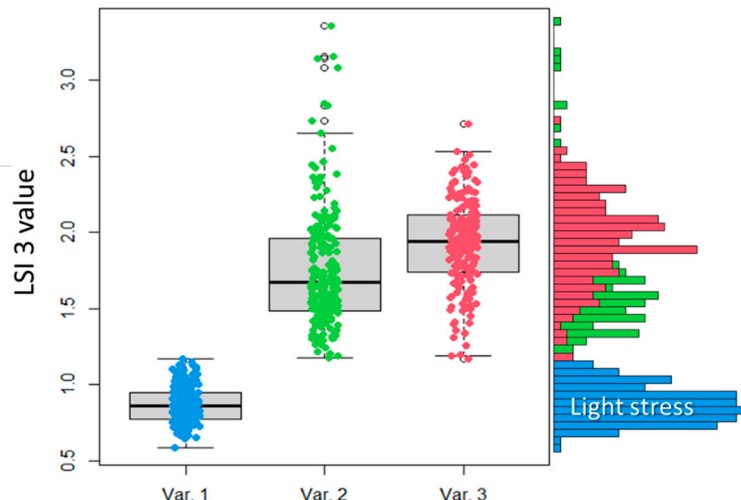

**Figure 12.** Distribution of the LSI 3 values of three LC 2 variants (Var. 1, Var. 2, and Var. 3) according to the results of HSI 3 and HSI 5 simultaneously.

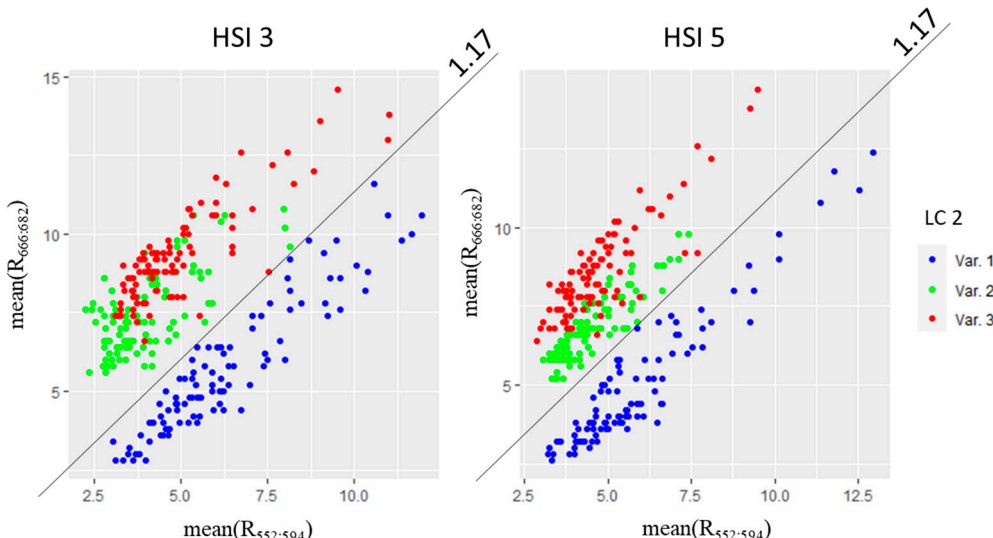

**Figure 13.** Distribution of mean($R_{666:682}$) against mean($R_{522:594}$) values of three variants (Var. 1, Var. 2, and Var. 3) for HSI 3 and HSI 5.

## 4. Discussion

Light stress [29–31], just like other types of stress [10–14], is known to be accompanied by a decrease in chlorophyll, which has two absorption maxima in the blue and red bands. On this basis, one would expect that, in response to light, the most pronounced changes in the reflectance spectra of the leaves would be an increase in the reflectance in these bands and possibly a slight increase in the reflectance in the green band. In addition, the far-red light range (680–740 nm) which corresponds to the chlorophyll fluorescence emission may be a stress-sensitive area, considering that it is inversely dependent on the photosynthesis rate [32]. However, studies have shown that light stress resulted in an increase in reflection in the yellow–green range and a decrease in reflection in the red range. In addition, the *t*-test and RF methods used in the experiments allowed for the of revealing three pairs of the most relevant ranges for light stress. They are 682–690 nm and 498–610 nm; 662–690 nm and 522–554 nm; and 666–682 nm and 522–594 nm. These ranges do not coincide with the sensitive ranges known for other types of stress. Thus, Navarro et al. [33] showed that ranges 492–504 and 540–568 nm of visible light; 712–720 nm of far-red light; and 855, 900–908, and 970 nm of near-infrared light had the best sensitivity to pathogen-induced

stress occurring in *Diplotaxis tenuifolia* plants. In several studies, other reflectance spectral ranges indicative of different types of stress (not related to photoinhibition) have been proposed, 850, 630, 535, and 465 nm [15]; 500–660 nm [34]; and 690 and 702 nm [9].

Light stress and light history had the most influence on the SB 690–700 nm and 730–750 nm [35], which is not consistent, but also not contradictory, with our data. The use of the three light-stress-sensitive ranges identified in our work makes it possible to increase the sensitivity and reliability of the indication of this type of stress.

The causes of the detected stress-induced increase in reflectance in the yellow–green range and the drop in reflectance in the red range are not clear. Possible reasons for this could be:

1.  Changes in pigment concentrations—decrease in carotenoids, which absorb well in the yellow–green range and do not absorb in the red range and increase in chlorophyll a, which, being a component of light-harvesting complexes (LHC) 1 and LHC 2, absorbs well in the 666–682 nm range, but does not absorb in the yellow-green range [36];
2.  Light-induced rearrangements of chloroplast localization, which can lead to shifts in light absorption by the pigments without changing their concentration [37].

However, irrespective of their cause (which is beyond the scope of this study), the observed shifts can serve as indicators of light stress and can be taken into account when calculating VIs. The light-stress index (LSI = mean($R_{666:682}$)/mean($R_{522:594}$) calculated on the basis of these ranges appeared to be the most sensitive to light stress compared to the SB and 80 VIs [28] selected for testing. If it is possible to develop and verify a scale for LSI, it would be possible to significantly simplify the identification of light stress.

In addition, the results of the study confirm the need for remote spectral monitoring of vegetation, taking into account its light history, which was previously pointed out by other authors [38–41].

In general, it can be said that the combination of the technical capabilities of a hyperspectral camera, with the method of machine learning [8,42,43] in the processing of hyperspectral data, opens up great prospects for the early diagnosis of plant stress.

## 5. Conclusions

The study performed in this work showed that the previous history of the illumination of leaves has a significant impact on their spectral characteristics. It has been established that two spectral ranges are significant for the identification of light stress. Intense artificial lighting leads to an increase in reflection in the range of 552–594 nm and a decrease in reflection in the range of 666–682 nm. Based on empirical data, the light stress index (LSI) = mean($R_{666:682}$)/mean($R_{552:594}$) was calculated and tested. LSI allows more accurate identification of the state of light stress compared to the SB and 80 VIs tested in the study. The LSI values are distributed according to the normal law, which makes it possible to use parametric statistics when working with it. It has been preliminarily established that the LSI threshold separating light stress from the normal state of the plant is 1.17. The results obtained indicate the possibility of identifying light stress in plants using spectral sensors that detect passive reflection. In addition, the results of the study confirm the need to take into account the light history in the remote spectral monitoring of vegetation.

The result of the study was reproduced in time in two repetitions. If this result is confirmed on other plant objects, then this may be of practical importance for monitoring the light stress of plants by means of remote sensing.

**Supplementary Materials:** https://www.mdpi.com/article/10.3390/agriengineering5040138/s1, Supplementary Table S1: Paired comparison using *t*-test; Supplementary Table S2: Spectral bands ranked by their contribution to reduction OOB estimate of error rate.

**Author Contributions:** Conceptualization, P.A.D., B.L.K. and V.S.L.; Data curation, B.L.K. and A.A.D.; Formal analysis, A.A.D.; Investigation, P.A.D. and A.A.D.; Methodology, P.A.D.; Project administration, P.A.D.; Resources, V.A.C. and T.V.V.; Software, A.A.D.; Writing—original draft, P.A.D. and B.L.K.; Writing—review and editing, P.A.D., B.L.K. and V.S.L. All authors have read and agreed to the published version of the manuscript.

**Funding:** The project was supported by the Russian Science Foundation under grant No. 22-14-00338, https://rscf.ru/project/22-14-00338/ (accessed on 1 September 2023), and performed at Southern Federal University (Rostov-on-Don, Russian Federation).

**Data Availability Statement:** Data are contained within the article and Supplementary Materials.

**Conflicts of Interest:** The authors declare no conflict of interest.

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
