# Peer review of "Indication of Light Stress in Ficus elastica Using Hyperspectral Imaging"

_agriengineering, doi:10.3390/agriengineering5040138_

Round 1

Reviewer 1 Report

Comments and Suggestions for Authors

The manuscript is devoted to the indication of light stress of plants using hyperspectral visualization.

The authors have obtained very interesting results.

Questions and comments on the text of the Manuscript:

- Section Introduction: Redundant Citation [7-11], [12-15], [20-26] it should be divided into several citations by reference 1-3.

- Methodology: 1) Why were Ficus elastica selected for the study, which are not agricultural plants and are not grown in the open ground?

2) How does the received radiation spectrum (Figure 2) correlate with solar radiation? Why was such a spectrum chosen, realized with the help of such radiation sources?

3) How were PPFD measured (lines 106, 107)?

Figure 4 is of poor quality.

Author Response

Dear Reviewer!

Authors are grateful to you for a careful and helpful positive analysis of our manuscript. Undoubtedly, due to your remarks/comments, the manuscript has been significantly improved. All remarks are reasonable, and we have corrected the MS in accordance with the comments and suggestions. The changes can be tracked in a track-changed version. The authors hope, the current version of the manuscript will meet your expectations regarding clarity and quality of presentation.

- Section Introduction: Redundant Citation [7-11], [12-15], [20-26] it should be divided into several citations by reference 1-3.

Response: Thank you for your comment. Fixed. The changes can be tracked in a track-changed version.

- Methodology: 1) Why were Ficus elastica selected for the study, which are not agricultural plants and are not grown in the open ground?

Response: Thank you for your comment. Ficus elastica is a convenient model object because it is a shade-tolerant short-day plant. Therefore, as we believe, it is the most sensitive to light stress. In doing so, we took into account that most tree species of boreal forests are shade-tolerant. In addition, species of the genus Citrus and Camellia sinensis are also evergreen shade-tolerant crops. Species of the genus Citrus and Camellia sinensis will be selected as objects of further research.

2) How does the received radiation spectrum (Figure 2) correlate with solar radiation? Why was such a spectrum chosen, realized with the help of such radiation sources?

Response: Thank you for your comment. A prerequisite for such studies is that the spectrum of artificial illumination should cover the entire spectral range of the hyperspectral camera. In addition, the intensity of radiation must be constant. And since the camera is calibrated before the study, the normalized spectra of artificial light sources and the sun will be the same.

3) How were PPFD measured (lines 106, 107)?

Response: Thank you for your comment. Levels of PPFD were measured using a PAR-sensor of Waltz Diving-PAM fluorometer. (added to the materials and methods section). The changes can be tracked in a track-changed version.

Figure 4 is of poor quality.

Response: Thank you for your comment. Fixed. The changes can be tracked in a track-changed version.

Reviewer 2 Report

Comments and Suggestions for Authors

This study is very interesting and it is recommended to revise it for publication.

This article compares the changes in spectral reflectance characteristics of elastic banyan leaves caused by 10 hours of exposure to light-suppressed white LED light and moderate natural light, as well as the adaptability of plants to darkness, and obtains many valuable results. 

The topic is original and it addresses a specific gap in the field.  The conclusions are consistent with the evidence and arguments presented and they do address the main question posed. The references are appropriate.

1. There are too many Figures in the article, and some images without scientific data can be considered as unnecessary, such as Figures 1, 2, 3, etc. 

2. Because there are too many Figures in this article, pay attention to writing standards, formatting, etc.

3. The format of the references in this article is not uniform. For example, some references have volumes and issues written, while others do not. Some page numbers are in pages and others are in PP.

Author Response

Dear Reviewer!

Authors are grateful to you for a careful and helpful positive analysis of our manuscript. Undoubtedly, due to your remarks/comments, the manuscript has been significantly improved. All remarks are reasonable, and we have corrected the MS in accordance with the comments and suggestions. The changes can be tracked in a track-changed version. The authors hope, the current version of the manuscript will meet your expectations regarding clarity and quality of presentation.

  1. There are too many Figures in the article, and some images without scientific data can be considered as unnecessary, such as Figures 1, 2, 3, etc.

Response: Thank you for your comment. Indeed, in relation to the volume of the text, there are quite a lot of figures. However, we assume that the article may be read not only by specialists from this field, so the figures may be useful for them to understand the article. Therefore, we would like to keep them.

  1. Because there are too many Figures in this article, pay attention to writing standards, formatting, etc.

Response: Thank you for your comment. We checked again with the journal's requirements for manuscript design.

  1. The format of the references in this article is not uniform. For example, some references have volumes and issues written, while others do not. Some page numbers are in pages and others are in PP.

Response: Thank you for your comment. Fixed. The changes can be tracked in a track-changed version.

Reviewer 3 Report

Comments and Suggestions for Authors

The manuscript is well-prepared. The work is very interesting and the presentation of the results is done precisely. However, in my opinion some improvements could me made. Also I have some questions:

Line 46 – how does this information apply to crops under cover? Plants in greenhouses or foil tunnels?

Line 84 – how long the plants grew in the described conditions before the analyzes were carried out for the first time?

Line 86 – what age?

Line 93 – remove the dot after the word ‘size’

Figure 2 – what is the unit on the y axis?

Fugure 4 – too small and illegible. Enlarge and describe as Figure 4A and Figure 4B, please

The background of Introduction is good. But are there any other methods to non-invasively measure the cause of stress in plants? What are their disadvantages if the Authors want to develop the discussed in manuscript technique?

The obtained and presented results should be compared with some others measurements. Will light stress identified in this way actually lead to a decrease in plant productivity  (the Authors mention that in the introduction section in lines 42-43)?. E.g. sometimes the treatment of plants with a stress factor causes the increased production of specific compounds. 

Did the authors consider conducting research using non-invasive methods to measure the efficiency of the photosynthesis process and photosynthetic apparatus? E.g. it’s possible damage or danger of permanent damage - does it occur?

Author Response

Dear Reviewer!

Authors are grateful to you for a careful and helpful positive analysis of our manuscript. Undoubtedly, due to your remarks/comments, the manuscript has been significantly improved. All remarks are reasonable, and we have corrected the MS in accordance with the comments and suggestions. The changes can be tracked in a track-changed version. The authors hope, the current version of the manuscript will meet your expectations regarding clarity and quality of presentation.

Line 46 – how does this information apply to crops under cover? Plants in greenhouses or foil tunnels?

Response: Thank you for your comment. In confined areas, e.g. in greenhouses, foil tunnels or open plantations, proximal spectral sensors can be used for stress recording. (added sentence)

Line 84 – how long the plants grew in the described conditions before the analyzes were carried out for the first time?

Response: Thank you for your comment. The plants were grown for 14 months. (added sentence). The changes can be tracked in a track-changed version.

Line 86 – what age?

Response: Thank you for your comment. All plants were 14 months old from the time of transfer to the soil from tissue culture.

Line 93 – remove the dot after the word ‘size’

Response: Thank you for your comment. Fixed.

Figure 2 – what is the unit on the y axis?

Response: Thank you for your comment. Arb. units (Arbitrary units). Fixed. The changes can be tracked in a track-changed version.

Fugure 4 – too small and illegible. Enlarge and describe as Figure 4A and Figure 4B, please

 Response: Thank you for your comment. Fixed. The changes can be tracked in a track-changed version.

 The background of Introduction is good. But are there any other methods to non-invasively measure the cause of stress in plants? What are their disadvantages if the Authors want to develop the discussed in manuscript technique?

Response: Thank you for your comment. Noninvasive methods of stress diagnostics may include the analysis of amplitude and kinetic characteristics of chlorophyll fluorescence. Technologies for monitoring solar-induced fluorescence (SIF) using satellites are currently under development (Zhang et al., 2019; Du et al., 2020). The method needs further development, and there are problems with the processing of the obtained signal. Passive remote sensing methods have been developed to a much greater extent, so we plan to develop the methodology in this direction.

Zhang, C., Atherton, J., Penuelas, J., Filella, I., Kolari, P., Aalto, J., Ruhanen, H., Bäck, J., Porcar-Castell, A. (2019). Do all chlorophyll fluorescence emission wavelengths capture the spring recovery of photosynthesis in boreal evergreen foliage?. Plant, Cell & Environment. 42. https://doi.org/10.1111/pce.13620.

Du, S.; Liu, L.; Liu, X.; Zhang, X.; Gao, X.; Wang, W. The Solar-Induced Chlorophyll Fluorescence Imaging Spectrometer (SIFIS) Onboard the First Terrestrial Ecosystem Carbon Inventory Satellite (TECIS-1): Specifications and Prospects. Sensors 2020, 20, 815. https://doi.org/10.3390/s20030815

The obtained and presented results should be compared with some others measurements. Will light stress identified in this way actually lead to a decrease in plant productivity  (the Authors mention that in the introduction section in lines 42-43)?. E.g. sometimes the treatment of plants with a stress factor causes the increased production of specific compounds. 

Response: Thank you for your comment. Indeed, small doses of stressors may have a stimulating effect. However, this is a subject for a separate experiment. In addition, we plan to test the results obtained under field conditions on agricultural crops.

Did the authors consider conducting research using non-invasive methods to measure the efficiency of the photosynthesis process and photosynthetic apparatus? E.g. it’s possible damage or danger of permanent damage - does it occur?

Response: Thank you for your comment. Very good point and a suggestion for our further research.
